# Critique Fine-Tuning: Learning to Critique is More Effective than Learning to Imitate

**Yubo Wang[1,3], Xiang Yue[2], Wenhu Chen[1,3]**
[1]Department of Computer Science, University of Waterloo
[2]Carnegie Mellon University, Pittsburgh [3]Vector Institute, Toronto

https://tiger-ai-lab.github.io/CritiqueFineTuning/

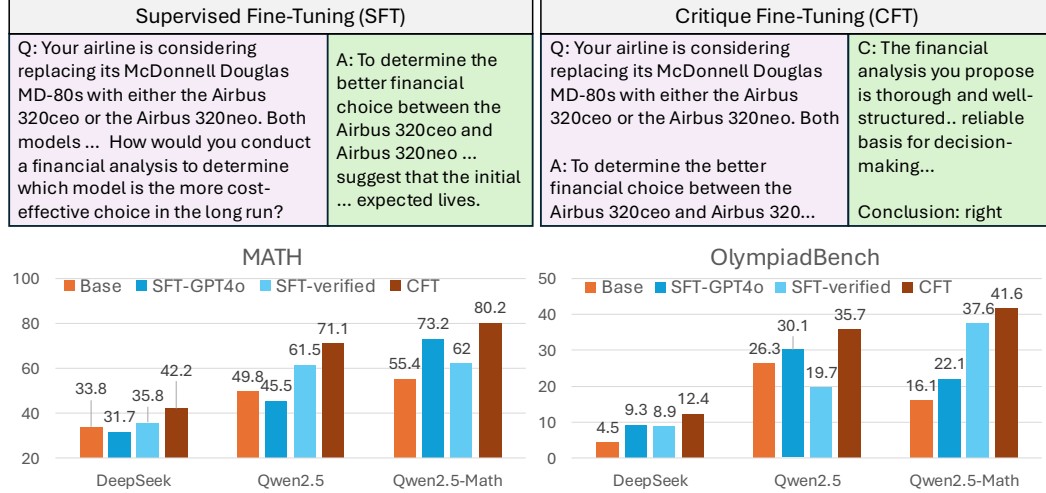

Figure 1: Comparison between CFT and SFT on 50K samples from WebInstruct (Yue et al., 2024b). SFT-verified means SFT training on the responses validated by GPT-4o, SFT-GPT4o means SFT training on the responses from GPT-4o. CFT is our approach, which trains on the critique provided by GPT-4o.

## Abstract

Supervised Fine-Tuning (SFT) is commonly used to train language models to imitate annotated responses for given instructions. In this paper, we propose Critique Fine-Tuning (CFT), a method more effective than SFT for reasoning tasks. Instead of simply imitating correct responses, CFT trains models to critique noisy responses, inspired by human learning processes that emphasize critical thinking, deeper analysis, and nuanced understanding—traits often overlooked by standard SFT. To validate the effectiveness of CFT, we construct multiple critique datasets (e.g., WebInstruct, MetaMath, NuminaMath), where GPT-4o serves as the teacher to generate critiques in the form of ([query; noisy response], critique). Experiments on these datasets demonstrate that CFT consistently outperforms SFT by 4–10% across six mathematical reasoning benchmarks, and is effective across different base models including Qwen2.5, Qwen2.5-Math, and DeepSeek-Math. Notably, our model Qwen2.5-Math-CFT only requires 1 hour of training on 8xH100 over the 50K examples, yet matches or outperforms strong competitors like Qwen2.5-Math-Instruct on most benchmarks, which use over 2M samples. Moreover, it matches the performance of SimpleRL, which is a DeepSeek-r1 replication trained with 140x more compute. Experiments on IF_Eval and MT-Bench further demonstrate that CFT can significantly enhance the model's general generation and instruction-following capabilities, outperforming the Qwen2.5-Math-Instruct by a large margin. Ablation studies show that CFT is robust to noisy response sources and teacher critique models. These findings highlight that CFT offers a more effective alternative to advance the reasoning of language models.

# 1 Introduction

Recently, large language models (LLMs) (Achiam et al., 2023; Team et al., 2023; Dubey et al., 2024) have shown unprecedented performance on real-world problems. One core techniques is supervised fine-tuning (SFT), which trains LLMs to follow natural language instructions (Wei et al., 2022; Ouyang et al., 2022; Sanh et al., 2022). **In the process of SFT, LLMs are forced to imitate the annotated responses**. Numerous efforts have been made to build high-quality SFT datasets using approaches like Self-Instruct (Wang et al., 2023b) and Evol-Instruct (Xu et al., 2024) to enhance LLMs' general instruction-following capabilities. More recently, works such as MAmmoTH (Yue et al., 2024a;b), MetaMath (Yu et al., 2024), and WizardCoder (Luo et al., 2024) have employed SFT to improve the targeted capabilities of LLMs in areas like mathematical reasoning, coding, and more. While these approaches have shown significant gains on weaker base models such as Mistral (Jiang et al., 2023) or LLaMA3 (Dubey et al., 2024), diminishing returns become evident as SFT dataset size and quality scale up. This limitation is particularly pronounced for already-powerful base models (non-SFTed), such as Qwen2.5-base (Yang et al., 2024a), Qwen2.5-Math-base (Yang et al., 2024b), or DeepSeek-Coder-V2-base (Guo et al., 2024), which have undergone extensive domain-adaptive pretraining on reasoning-focused corpora comprising hundreds of billions of tokens. Our experiments in section 3 reveal that applying SFT to these models can even degrade performance without stringent quality control.

In this paper, we propose a new learning framework called Critique Fine-Tuning (CFT) as an alternative to traditional SFT. Inspired by human learning—where critical thinking and constructive feedback are vital for improvement—we shift the focus from imitation to critique-based learning. When humans learn, they do not merely replicate provided answers but analyze, critique, and refine them. Similarly, in CFT, the model learns to provide critiques for noisy responses, identify flaws, suggest improvements, and verify correctness. **Formally, CFT involves training the model to critique a given query-response pair, maximizing the likelihood** $P(c|[x;y])$**, where** $c$ **is the annotated critique for a query-response pair** $[x;y]$. A detailed visualization of CFT is presented in Figure 1.

To validate CFT's effectiveness, we designed a series of experiments using multiple critique datasets, including WebInstruct (Yue et al., 2024b), MetaMathQA (Yu et al., 2024), and NuminaMath (Li et al., 2024b), with critiques synthesized by advanced models such as GPT-4o (Achiam et al., 2023). We applied CFT to strong 7B base language models (non-instruction-tuned), including DeepSeekMath-base (Shao et al., 2024), Qwen2.5 (Yang et al., 2024a), and Qwen2.5-Math (Yang et al., 2024b). These models were compared against strong SFT-trained variants, such as WebInstruct-verified (SFT on GPT-4o-verified responses) and WebInstruct-GPT4o (SFT directly on GPT-4o-generated responses). When evaluated on six math benchmarks, including MATH and AIME24, CFT-trained models consistently outperform the best SFT-trained models by an average of 4–10 absolute points.

We further expanded the evaluation to broader STEM benchmarks, including GPQA (Rein et al., 2023), TheoremQA (Chen et al., 2023), and MMLU-Pro (Wang et al., 2024b). Our results show that the best CFT-trained model, Qwen2.5-Math-CFT, trained on 50K examples, outperformed strong competitors like AceMath (Liu et al., 2024) and Qwen2.5-Math-Instruct (Yang et al., 2024b), which were trained on over 2M examples. We also compare Qwen2.5-Math-CFT with SimpleRL (Zeng et al., 2025), an open replication of DeepSeek-R1 (Guo et al., 2025) trained with 140x more compute (1152 vs 8 H100 hours). Results show that Qwen2.5-Math-CFT reaches the same average performance across 5 math benchmarks, highlighting the efficiency and effectiveness of CFT for reasoning-focused tasks.

To understand the models' general instruction following abilities, we also evaluate our Qwen2.5-Math-CFT on MT-Bench (Zheng et al., 2023) and IF_Eval (Zhou et al., 2023). We show that our method also enhances general instruction-following abilities. It outperforms our SFTed version and the official Qwen2.5-Math-Instruct by a notable margin.

To better understand different factors of CFT, we conducted comprehensive ablation studies:

**1. Robustness to dataset sources:** Comparing WebInstruct (Yue et al., 2024b) against Meta-MathQA (Yu et al., 2024) and NuminaMath (Li et al., 2024b), we observed that WebInstruct

provided a slight advantage (3%+) due to its diversity and broader topic coverage.
**2. Robustness to noisy response sources:** Experiments with original noisy responses and Qwen2.5-base responses critiqued by GPT-4o showed negligible performance differences.
**3. Flexibility to the teacher:** Using a weaker critique dataset synthesized by GPT-4o-mini still yielded notable improvements over SFT, despite a 4% overall score drop.
**4. Controlling for token length:** Even controlling token length, shorter critique examples significantly outperformed standard SFT (55.2% vs. 50.4%), confirming improvements stem from critique-based training rather than increased sequence length.

Our approach not only demonstrates strong improvement on reasoning-focused tasks, but also exhibits notable improvement over general-purpose instruction following tasks. These evidence has shown great potential of CFT to replace SFT in language model training.

## 2   Method & Dataset

To validate the effectiveness of CFT, we construct several fine-tuning datasets. Most of our experiments are based on WebInstruct, an instruction dataset collected from online educational resources and quiz websites. The dataset undergoes synthetic processing in its pipeline using LLMs to improve solution quality and format consistency.

### 2.1   WebInstruct

WebInstruct spans a wide range of topics, including Mathematics (65%), Physics (8%), Chemistry (4%), Business (10%), Humanities (4%), and more. Unlike other datasets, which are primarily derived from math contests and competitions, WebInstruct offers broader topic coverage. The responses in WebInstruct are extracted and refined by large language models such as Qwen-72B (Bai et al., 2023) and Mixtral (Jiang et al., 2024), making them highly prone to noise due to the lack of verification or quality control.

We curate the following subsets from WebInstruct:

- WebInstruct-SFT: A 50K subset directly sampled from the original WebInstruct dataset. This subset has a very high error ratio (over 50%).
- WebInstruct-verified: We prompt GPT-4o-1120 to judge the correctness of original WebInstruct answers. We retained the top 50K samples as "verified" SFT data.
- WebInstruct-GPT-4o: A 50K subset that reuses questions from WebInstruct-SFT but replaces the answers with those generated by GPT-4o-1120.
- WebInstruct-CFT (Ours): A 50K subset derived from WebInstruct-SFT, where GPT-4o-1120 provides detailed critiques of the original responses. Approximately 56% of the responses in this subset are judged as "correct" while the rest are considered "wrong". Despite containing some critique errors introduced by GPT-4o, this dataset is comparable in quality to WebInstruct-GPT-4o.

We compare our CFT datasets with existing SFT datasets in Table 1. As shown, our datasets cover a broader range of topics while being significantly smaller in size, highlighting their efficiency in boosting LLMs' reasoning abilities.

### 2.2   MetaMath & NuminaMath

In addition to WebInstruct, we synthesized critiques for other datasets, including Meta-MathQA and NuminaMath. From each dataset, we randomly sampled 50K examples and used GPT-4o to critique the original responses. We then applied CFT to these datasets to demonstrate the generalizability of our approach across other datasets.

### 2.3   Training Objective

Our training objective is straightforward. We concatenate question $x$ and noisy response $y$ as input, then optimize the model parameters to generate critique $c$. Formally, the training loss is: $argmax_\theta \log P(c|[x;y];\theta)$ where $\theta$ is the parameters of the language model.

| Dataset | Size | Source or Seed | Discipline |
|---------|------|----------------|------------|
| Supervised Fine-Tuning Data | | | |
| WizardMath (Luo et al., 2023) | 96K | GSM8K, MATH | Math |
| MathInstruct (Yue et al., 2024a) | 260K | GSM8K, MATH, etc | Math |
| MetaMathQA (Yu et al., 2024) | 395K | GSM8K, MATH | Math |
| XwinMath (Li et al., 2024a) | 1.4M | GSM8K, MATH | Math |
| OrcaMath (Mitra et al., 2024) | 200K | GSM8K | Math |
| NuminaMath (Li et al., 2024b) | 860K | GSM8K, MATH, AIME | Math |
| AceMath (Liu et al., 2024) | 1.6M | GSM8K, MATH, AIME | Math |
| OpenMath-2 (Toshniwal et al., 2024) | 14M | GSM8K, MATH | Math |
| Critique Fine-Tuning Data (Ours) | | | |
| CFT | 50K | WebInstruct | STEM |

Table 1: The comparison of CFT and SFT datasets.

# 3 Experiments

In this section, we will detail our experiments.

| Model | Method | MATH | Minerva-Math | GSM8K | OlympiadBench | AIME24 | AMC23 | AVG |
|-------|--------|------|--------------|-------|---------------|--------|-------|-----|
| | Base | 33.8 | 9.2 | 64.3 | 4.5 | 0.0 | 10.0 | 20.3 |
| DeepSeek-Math-7B | WebInstruct-SFT | 26.3 | 12.1 | 34.7 | 6.2 | 0.0 | 17.5 | 16.1 |
| | WebInstruct-verified-SFT | 35.8 | 10.7 | 67.5 | 9.3 | 0.0 | 7.5 | 21.8 |
| | WebInstruct-GPT4o-SFT | 31.7 | 11.8 | 70.9 | 8.9 | 3.3 | 17.5 | 24.0 |
| | WebInstruct-CFT | 42.2 | 12.5 | 74.5 | 12.4 | 3.3 | 20.0 | 27.5 |
| | $\Delta$ = CFT- SFT$_{best}$ | 6.4 | 0.4 | 3.6 | 3.1 | 0.0 | 2.5 | 3.5 |
| | Base | 49.8 | 15.1 | 85.4 | 26.3 | 10.0 | 37.5 | 37.4 |
| Qwen2.5-7B | WebInstruct-SFT | 30.8 | 6.6 | 59.5 | 5.8 | 3.3 | 15.0 | 20.2 |
| | WebInstruct-verified-SFT | 61.5 | 16.2 | 70.8 | 30.1 | 13.3 | 37.5 | 38.2 |
| | WebInstruct-GPT4o-SFT | 45.5 | 18.4 | 77.4 | 19.7 | 10.0 | 50.0 | 36.8 |
| | WebInstruct-CFT | 71.1 | 27.9 | 88.8 | 35.7 | 13.3 | 55.0 | 48.6 |
| | $\Delta$ = CFT- SFT$_{best}$ | 9.6 | 9.5 | 11.4 | 5.6 | 0.0 | 5.0 | 10.4 |
| | Base | 55.4 | 13.6 | 91.6 | 16.1 | 10.0 | 40.0 | 37.8 |
| Qwen2.5-Math-7B | WebInstruct-SFT | 59.0 | 13.2 | 77.4 | 19.9 | 3.3 | 37.5 | 35.1 |
| | WebInstruct-verified-SFT | 62.0 | 12.5 | 78.8 | 22.1 | 16.7 | 50.0 | 40.4 |
| | WebInstruct-GPT4o-SFT | 73.2 | 25.7 | 90.0 | 37.6 | 13.3 | 62.5 | 50.4 |
| | WebInstruct-CFT | 80.2 | 42.3 | 90.9 | 41.6 | 20.0 | 67.5 | 57.1 |
| | $\Delta$ = CFT- SFT$_{best}$ | 7.0 | 16.6 | 0.9 | 4.0 | 3.3 | 5.0 | 6.7 |

Table 2: Performance comparison of SFT and CFT on different base models. All the experiments are trained with WebInstruct subset. We select the checkpoint with highest validation score and report their results.

## 3.1 Experimental Setup

**Evaluation Datasets** We evaluate our method on a wide range of mathematical reasoning benchmarks. For standard mathematical reasoning, we evaluate MATH (Hendrycks et al., 2021), Minerva-Math (Lewkowycz et al., 2022) and GSM8K (Cobbe et al., 2021). To assess performance on more challenging competition-level mathematics, we incorporate AIME 2024, AMC 2023, and OlympiadBench (He et al., 2024) containing various difficulty levels of Mathematical Olympiad problems. We further extend our evaluation to broader STEM reasoning capabilities through TheoremQA (Chen et al., 2023) for mathematical theorem understanding, MMLU-Pro (Wang et al., 2024b) covering physics, chemistry, mathematics, etc., and GPQA (Rein et al., 2023) for complex problems requiring scientific reasoning.

**Training Details** We evaluate three different SFT training settings and one CFT training setting in our experiments. For the SFT training settings, we explore: (1) SFT: direct training on original dataset, (2) SFT-verified: training on responses validated by GPT-4o, and (3) SFT-GPT-4o: training on responses generated by GPT-4o. For CFT, we train the model using our curated CFT datasets as described in section 2. We use MATH-500 (Lightman et al., 2023b) as our validation set and select the best-performing checkpoint after training on the

| Model | #Data | MATH | GPQA | TheoremQA | MMLU-Pro | OlympiadBench | AIME24 | AMC23 | AVG |
|---|---|---|---|---|---|---|---|---|---|
| *Frontier Models* | | | | | | | | | |
| GPT-4o (2024-08-06) | - | 81.1 | 51.6 | 54.7 | 74.7 | 43.3 | 9.3 | 47.5 | 51.7 |
| GPT-o1-mini | - | 90.0 | 60.0 | 57.2 | 80.3 | 65.3 | 56.7 | 95.0 | 72.1 |
| *Other Open-sourced Reasoning LLMs* | | | | | | | | | |
| Deepseek-Math-7B-Instruct | - | 44.3 | 31.8 | 23.7 | 35.3 | 13.6 | 3.3 | 15.0 | 23.9 |
| Mathstral-7B-v0.1 | - | 56.6 | 32.2 | 28.4 | 42.5 | 21.5 | 6.7 | 42.4 | 32.9 |
| NuminaMath-7B-CoT | - | 55.2 | 30.6 | 28.6 | 38.6 | 19.9 | 6.7 | 30.0 | 29.9 |
| Llama-3.1-8B-Instruct | - | 51.9 | 30.4 | 30.3 | 48.3 | 14.4 | 6.7 | 30.0 | 30.3 |
| Llama-3.1-70B-Instruct | - | 65.7 | 42.2 | 51.3 | 62.8 | 14.4 | 16.7 | 30.0 | 40.4 |
| NuminaMath-72B-CoT | - | 68.0 | 35.3 | 24.9 | 55.0 | 35.0 | 3.3 | 52.5 | 39.1 |
| Qwen2.5-Math-72B-Instruct | - | 85.9 | 49.0 | 50.3 | 60.3 | 49.0 | 30.0 | 70.0 | 56.4 |
| *Initialized from Qwen2.5-Math-7B-Base* | | | | | | | | | |
| Qwen2.5-Math-Base | 0 | 55.4 | 31.0 | 37.4 | 39.3 | 16.1 | 10.0 | 40.0 | 32.7 |
| Eurus-2-SFT | 230 K | 62.4 | 32.1 | 38.0 | 44.2 | 29.8 | 3.3 | 30.1 | 34.3 |
| rStar-Math@Greedy | 747 K | 78.4 | - | - | - | **47.1** | **26.7** | 47.5 | - |
| AceMath-Qwen2.5-Math | 2.3 M | 83.1 | 26.1 | 24.6 | **48.1** | 42.2 | 16.7 | 60.0 | 43.0 |
| Qwen2.5-Math-7B-Instruct | 2.5 M | **83.6** | 31.1 | 37.0 | 39.5 | 41.6 | 16.7 | 62.5 | 44.6 |
| Qwen2.5-Math-7B-CFT | 50 K | 80.2 | **39.4** | **40.4** | 47.5 | 41.6 | 20.0 | **67.5** | 48.1 |

Table 3: Performance comparison of our models vs. other reasoning-specialized models. #Data denotes total training set size. We select the checkpoint with highest validation score.

entire dataset for 1 epoch. We maintain consistent hyperparameters across all experiments with a learning rate of 5e-6, a cosine decay learning schedule with a warm-up ratio of 0.1, and a global batch size of 512.

## 3.2 Main Results (CFT vs. SFT)

To evaluate the effectiveness of CFT, we compare it with various SFT methods on three 7B-scale base models using mathematical reasoning benchmarks. Table 2 presents the results across different base models and methods. Our key findings are as follows:

**Base Model Selection** We experiment with three 7B-scale base models: DeepSeek-Math-7B, Qwen2.5-7B, and Qwen2.5-Math-7B. Results show that Qwen2.5-Math-7B serves as a stronger foundation, with its base version achieving 37.8% average accuracy across benchmarks. When enhanced with CFT, it achieves the best performance with 57.1% average accuracy.

**Performance Gains** CFT consistently outperforms all SFT baselines across different models. On DeepSeek-Math-7B, it achieves a 3.5% absolute improvement over the SFT-GPT4o. On Qwen2.5-7B, it demonstrates a substantial 10.4% improvement over the SFT-verified. On Qwen2.5-Math-7B, it surpasses the strong GPT-4o SFT baseline by 6.7% over SFT-GPT4o.

## 3.3 More Results (CFT Models vs. Existing Models)

In Table 3, we compare our best CFT-models with other competitive models with different scales. We expanded the evaluation benchmarks to cover broader STEM topics.

Our Qwen2.5-Math-7B-CFT achieves the highest average performance (48.1%) among 7B-scale models while using significantly less training data (50K samples). Specifically:

- It substantially outperforms specialized math models such as Deepseek-Math-7B-Instruct (23.9%), Mathstral-7B (32.9%), and NuminaMath-7B-CoT (29.9%).
- This strong performance is achieved with remarkably less training data - only 50K samples compared to AceMath-Qwen2.5-Math (2.3M samples) and Qwen2.5-Math-7B-Instruct (2.5M samples), demonstrating the superior data efficiency of our approach. In addition, our method outperforms Eurus-2-SFT (230K) Cui et al. (2025) and rStar-Math@Greedy (747K) Guan et al. (2025) on most tasks.

Despite being smaller in scale, our Qwen2.5-Math-7B-CFT demonstrates strong performance compared to larger models. With only 7B parameters, it achieves better average performance (48.1%) than Llama-3.1-70B-Instruct (40.4%) and NuminaMath-72B-CoT (39.1%). When

compared to Qwen2.5-Math-72B-Instruct (56.4%), our model shows competitive results on several benchmarks (e.g., 67.5% vs 70.0% on AMC23) despite using only one-tenth of the parameters and less training data. While frontier closed models like GPT-4o still maintain a performance lead, our results demonstrate that efficient training strategies can help smaller models achieve strong performance with fewer resources.

## 3.4 Comparison with RL-based Method

| Model | Data Size | GPU Hours | MATH-500 | Minerva-Math | OlympiadBench | AIME24 | AMC23 | AVG |
|---|---|---|---|---|---|---|---|---|
| SimpleRL-Zero | 8K×12 | 1152 | 77.2 | 33.5 | 37.9 | 33.3 | 62.5 | 48.9 |
| SimpleRL | 8K+8K×12 | 1152 | 82.4 | 39.7 | 43.3 | 26.7 | 62.5 | **50.9** |
| CFT | 50K | 8 | 79.6 | 42.3 | 41.6 | 20.0 | 67.5 | 50.2 |

Table 4: Performance comparison with RL-driven methods.

Recently, researchers have shown that reinforcement learning can significantly boost the reasoning capabilities of large language models. Here, we compare with SimpleRL (Zeng et al., 2025), which is an open replication of DeepSeek-R1 (Guo et al., 2025). We consider pure RL-based training (SimpleRL-Zero) and Distill+RL-based training (SimpleRL) as our competitors, both require 32xH100 to train for 1.5 days. In contrast, our method only requires 8xH100 to train for 1 hour. Furthermore, our approach does not requirea long decoding length leading to higher efficiency.

As shown in Table 4, CFT can improve Qwen2.5-Math-7B-base to the same level as SimpleRL. On several benchmarks like AMC23 and Minverva-Math, CFT can outperform both SimpleRL significantly. The biggest difference is AIME24, which only contains a test set of 30 questions. The accuracy is heavily impacted by the randomness.

## 3.5 Ablation Studies

To understand the impact of different factors in CFT, we conduct several ablation studies:

| Task | MetaMathQA | | NuminaMath | | WebInstruct | |
|---|---|---|---|---|---|---|
| | SFT | CFT | SFT | CFT | SFT | CFT |
| MATH | 57.5 | 74.4 | 70.8 | 74.2 | 59.0 | 80.2 |
| Minerva-Math | 23.9 | 42.3 | 28.3 | 32.8 | 13.2 | 42.3 |
| GSM8K | 79.5 | 85.7 | 88.3 | 89.1 | 77.4 | 90.9 |
| OlympiadBench | 20.0 | 36.4 | 36.3 | 37.2 | 19.9 | 41.6 |
| AIME24 | 6.7 | 23.3 | 10.0 | 23.3 | 3.3 | 20.0 |
| AMC23 | 37.5 | 57.5 | 50.0 | 62.5 | 37.5 | 67.5 |
| AVG | 37.5 | 53.3 | 47.3 | 53.2 | 35.1 | 57.1 |

Table 5: Performance comparison of SFT and CFT with different training datasets on Qwen2.5-Math-7B.

| Task | Base | Self-generated | Other-generated |
|---|---|---|---|
| MATH | 55.4 | 78.2 | 80.2 |
| Minerva-Math | 13.6 | 33.1 | 42.3 |
| GSM8K | 91.6 | 92.4 | 90.9 |
| OlympiadBench | 16.1 | 42.5 | 41.6 |
| AIME24 | 10.0 | 16.7 | 20.0 |
| AMC23 | 40.0 | 67.5 | 67.5 |
| AVG | 37.8 | 55.1 | 57.1 |

Table 6: Comparison between self-generated (by Qwen2.5-Math-7B) and original solutions (from WebInstruct) for CFT training.

| Task | SFT | GPT-4o-mini-CFT | GPT-4o-1120-CFT |
|---|---|---|---|
| MATH | 62.0 | 73.9 | 80.2 |
| Minerva-Math | 12.5 | 36.4 | 42.3 |
| GSM8K | 78.8 | 84.5 | 90.9 |
| OlympiadBench | 22.1 | 35.1 | 41.6 |
| AIME24 | 16.7 | 20.0 | 20.0 |
| AMC23 | 50.0 | 62.5 | 67.5 |
| AVG | 40.4 | 52.0 | 57.1 |

Table 7: Performance comparison of CFT using different teacher critique models (GPT-4o and mini) on Qwen2.5-Math-7B.

| Model | IF_Eval | | MT-Bench |
|---|---|---|---|
| | strict | loose | |
| Qwen2.5-Math-7B | 0.266 | 0.291 | 4.79 |
| Qwen2.5-Math-7B-Instruct | 0.333 | 0.345 | 5.49 |
| Qwen2.5-Math-7B-SFT | 0.315 | 0.330 | 5.23 |
| Qwen2.5-Math-7B-verified-SFT | 0.328 | 0.341 | 5.41 |
| Qwen2.5-Math-7B-GPT4o-SFT | 0.325 | 0.343 | 5.38 |
| Qwen2.5-Math-7B-CFT | 0.335 | 0.362 | 6.49 |

Table 8: Performance comparison across different models on Instruction Following (IF_Eval, instruction-level) and general instruction alignment (MT-Bench).

**Dataset Source** We ablate the impact of different training datasets on model performance. As shown in Table 5, when trained with SFT, both MetaMathQA and NuminaMath achieve better performance than WebInstruct (47.3% and 37.5% vs. 35.1% on average), indicating their higher data quality. However, when trained with CFT, WebInstruct surprisingly achieves the best performance (57.1%), outperforming both MetaMathQA and NuminaMath. This suggests that the effectiveness of CFT is not solely determined by the quality of solution data. Instead, by learning to identify and critique incorrect solutions, the model can develop stronger mathematical reasoning capabilities even from imperfect demonstrations, highlighting the robustness and effectiveness of our critique-based learning approach.

**Response Source** We compare two sources of solutions for CFT training: solutions generated by Qwen2.5-Math-7B itself and original solutions from the WebInstruct dataset. Table 6 shows that using original solutions achieves comparable performance (57.1% vs. 55.1% on average), with some variation across different benchmarks. The improvement is more noticeable on challenging datasets like Minerva-Math (9.2% increase). These results demonstrate that CFT is robust to different solution sources and can effectively learn from both model-generated and original solutions from the WebInstruct dataset.

**Teacher Critique Model** To understand the impact of critique model quality on CFT, we compare the performance when using GPT-4o-mini and GPT-4o-1120 as critique models in Table 7. First, we observe that even with a relatively modest critique model GPT-4o-mini, CFT significantly outperforms SFT-verified baseline (52.0% vs. 40.4% on average), with substantial improvements on MATH (11.9% increase) and Minerva-Math (23.9% increase). This demonstrates the effectiveness of CFT without requiring an extremely powerful critique model. Furthermore, using a stronger critique model GPT-4o-1120 leads to even better performance across all benchmarks (57.1% on average), with notable gains on GSM8K (6.4% increase) and OlympiadBench (6.5% increase). These results confirm that while CFT is effective with modest critique models, stronger critique models can provide more accurate and insightful feedback, leading to better mathematical reasoning capabilities. In the future, we plan to leverage o1 or even o3 as teacher critique model to understand the potential.

| Model | MATH | Minerva-Math | GSM8K | OlympiadBench | AIME24 | AMC23 | AVG |
|---|---|---|---|---|---|---|---|
| Qwen2.5-Math-7B-CFT | **80.2** | **42.3** | **90.9** | **41.6** | **20.0** | **67.5** | **57.1** |
| Mix 50K CFT + 50K AceMath SFT | 78.8 | 40.8 | 89.9 | 40.3 | 16.7 | 62.5 | 54.8 |
| CFT + 50K AceMath SFT (two-stage) | 76.2 | 35.4 | 88.2 | 42.5 | 10.0 | 57.5 | 51.6 |
| AceMath SFT (50K) | 73.7 | 33.1 | 90.2 | 35.1 | 13.3 | 52.5 | 49.7 |

Table 9: Comparison of strategies for combining CFT with high-quality SFT data

| Model | MATH | Minerva-Math | GSM8K | OlympiadBench | AIME24 | AMC23 | AVG |
|---|---|---|---|---|---|---|---|
| WebInstruct-SFT | 59.0 | 13.2 | 77.4 | 19.9 | 3.3 | 37.5 | 35.1 |
| WebInstruct-verified-SFT | 62.0 | 12.5 | 78.8 | 22.1 | 16.7 | 50.0 | 40.4 |
| WebInstruct-GPT4o-SFT | 73.2 | 25.7 | 90.0 | 37.6 | 13.3 | 62.5 | 50.4 |
| WebInstruct-CFT | **80.2** | **42.3** | **90.9** | **41.6** | **20.0** | **67.5** | **57.1** |
| WebInstruct-CFT-Short | 78.4 | 40.4 | 90.4 | 40.1 | 16.7 | 65.0 | 55.2 |

Table 10: Ablation on controlling token length (CFT vs. SFT)

**Combination with High-Quality SFT Data** We investigate whether combining CFT with high-quality SFT datasets can further improve performance. As shown in Table 9, mixing AceMath SFT data or applying it in a two-stage training setting slightly degrades performance compared to pure CFT training (54.8% and 51.6% vs. 57.1% on average, respectively). This indicates that incorporating standard supervised instruction data may conflict with critique-based training objectives. Therefore, purely learning to critique incorrect solutions is more effective for reasoning tasks than jointly learning to imitate high-quality solutions.

**Controlling for Token Length** Since critique-based training typically involves longer sequences, one might suspect this as the primary reason for its superior performance. To address this concern, we select WebInstruct data with token lengths similar to standard SFT training data and perform critique fine-tuning. The results in Table 10 show that even when controlling token lengths, shorter-length critique data still significantly outperforms SFT

baselines (55.2% vs. 50.4% on average). This confirms that the advantage of CFT comes from the critique-based learning mechanism rather than merely increased token length.

**General Generation and Instruction-Following Capability**

To assess whether CFT affects the general generation and instruction-following abilities, we evaluate the models on MT-Bench (Zheng et al., 2023), a benchmark for general instruction alignment, and IF_Eval (Zhou et al., 2023), a comprehensive instruction-following evaluation suite. As shown in Table 8, Qwen2.5-Math-7B-CFT achieves a score of 6.49 on MT-Bench, significantly improving over the base Qwen2.5-Math-7B (4.79) and instruction-tuned Qwen2.5-Math-7B-Instruct (5.49). It also outperforms various SFT approaches, including standard SFT (5.23), verified-SFT (5.41), and GPT4o-SFT (5.38). On IF_Eval, Qwen2.5-Math-7B-CFT demonstrates superior instruction-following ability with the highest scores in both strict mode (0.335) and loose mode (0.362), outperforming all other variants including the instruction-tuned model. These comprehensive improvements across different benchmarks demonstrate that critique-based training not only enhances mathematical reasoning but also improves general instruction-following capabilities and text generation quality. This indicates that the critique mechanism helps the model develop more robust general capabilities even when trained primarily on mathematical content, suggesting positive transfer between specialized and general abilities.

# 4 Limitations

**The Noisy Critique Data** Our ablation study indicates that the quality of critique feedback notably influences the effectiveness of CFT. A manual inspection of 50 critique samples generated by GPT-4o-1120 on WebInstruct reveals that roughly 20% contain inaccuracies, such as misjudging correct steps, missing errors, or providing imprecise explanations (see Appendix A.3 for examples). This highlights the importance of using higher-quality critique data to further enhance CFT. Future research could explore automated critique verification methods or develop curated datasets with human-verified critiques to improve mathematical reasoning capabilities.

**Limitations of Self-Critique**

We explored incorporating self-critique mechanisms into our framework, where the model critiques and iteratively refines its own outputs. However, we found these approaches consistently underperformed compared to direct inference. In particular, self-critique suffered from inconsistent critique standards, where the model either overlooked genuine errors or mistakenly flagged correct solutions as incorrect. Additionally, increasing sampling temperatures—necessary to maintain diversity and avoid repetition during iterative refinement—introduced instability, further degrading performance. Due to these challenges, our final CFT implementation relies on direct inference without self-critique. For detailed experimental results, specific methodological comparisons, and further analysis of observed issues, please refer to Appendix A.4.

# 5 Related Work

## 5.1 Instruction Tuning

Instruction tuning is one of the most crucial part of aligning pre-trained language models with human expectations. The current instruction-tuning datasets are either based on (1) human annotation: such as FLAN (Wei et al., 2022), T0 (Sanh et al., 2022), SuperNI (Wang et al., 2022), which compiles large instruction-tuning datasets from existing human-labeled datasets; and (2) model synthesis: such as Self-Instruct (Wang et al., 2023b), WizardLM (Xu et al., 2024), WildChat (Zhao et al., 2024), which creates instruction-tuning datasets by synthesizing from powerful LLMs (Achiam et al., 2023). Both types of instruction datasets have shown great performance improvement of LMs on general evaluation tasks. More recently, Tulu (Wang et al., 2023a) and Tulu-3 (Lambert et al., 2024) have explored how to combine existing post-training data and algorithms to maximize LMs' performance.

## 5.2 Mathematical Instruction Tuning

Taking this further, math-instructed models have been developed to advance LLM performance in the mathematical domain (Luo et al., 2023; Yue et al., 2024a; Yu et al., 2024; Li et al., 2024a; Mitra et al., 2024; Li et al., 2024b). Recently, there has been a wave to scale up the math instruction dataset to millions of examples like MAmmoTH2 (Yue et al., 2024b), Open-MathInstruct (Toshniwal et al., 2024), and AceMath (Liu et al., 2024) and Qwen2.5-Math-Instruct (Yang et al., 2024b). These methods have shown tremendous performance gains on math reasoning datasets. However, we also observe diminishing marginal gain by further scaling the instruction data up, suggesting that a more efficient training algorithm is needed. In this paper, we aim to challenge SFT and propose a much more efficient learning algorithm CFT and show similar performance with only 1-10% of SFT data.

## 5.3 Critique Learning

Teaching AIs to critique has been a long standing goal in the pursuit of AGI.

**Self-Correction** The concept of 'self-correction' has been emerged as a promising direction in LLMs since 2023. There has been a line of work (Madaan et al., 2024; Welleck et al., 2023; Shinn et al., 2024; Bai et al., 2022; Ganguli et al., 2023; Gou et al., 2023) aiming to use feedback from the model itself to further improve its performance. However, a later work (Huang et al., 2023; Valmeekam et al., 2023) revealed that the self-correction in reasoning is not quite reliable. More recently, with the rise of GPT-o1 (Jaech et al., 2024), LLM self-correction has again demonstrated its potential to improve LLMs' own reasoning capabilities.

**Critique Model** A critique model differs from self-correction in that it employs a specialized model to provide feedback to another model during generation. In mathematical reasoning, critique models often take the form of reward models. Recent work has explored both outcome-based reward models (Uesato et al., 2022; Yang et al., 2024b) and process-based reward models (Wang et al., 2024a; Lightman et al., 2023a; Yuan et al., 2024) to enhance reasoning capabilities of language models. However, these approaches typically focus on directly predicting reward scores without explicitly providing intermediate reasoning or explanations. The most similar prior work to ours is critique-out-loud (Ankner et al.), which functions solely as a reward estimator rather than directly guiding the generation process.

In contrast, our proposed approach differs substantially from these existing methods. We leverage critique feedback explicitly as a training objective to encourage deeper understanding and reasoning about the given problems. At inference time, the trained model generates responses directly, without involving any explicit critique or iterative refinement steps.

## 6 Conclusion

In this paper, we introduced Critique Fine-Tuning (CFT), a novel paradigm that trains language models to critique and analyze responses rather than imitating them as in traditional SFT. Experiments demonstrated that CFT consistently outperforms SFT by 4–10% on mathematical reasoning benchmarks, achieves comparable performance to resource-intensive RL methods using significantly fewer training examples (50K vs. 2M+) and compute (8 H100 GPU-hours), and generalizes effectively to broader STEM domains. Interestingly, even without explicit instruction tuning via traditional SFT or RL, CFT-trained models inherently exhibit strong instruction-following capabilities, challenging conventional assumptions regarding the necessity of imitation-based training for instruction-following tasks. These findings suggest that explicitly teaching models to identify and critique incorrect reasoning can significantly enhance their reasoning and generalization capabilities. Future research directions include improving the quality and coverage of critique data, enabling models to perform self-critique for continual self-improvement, combining CFT with complementary training paradigms like RL, extending the approach to multi-modal scenarios, and further investigating its theoretical foundations. Overall, we believe CFT represents a promising step forward in making language model training more efficient, robust, and effective, potentially reducing the computational and data requirements for developing high-performing models while substantially improving their reasoning and instruction-following abilities.

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

# A Appendix

## A.1 GPT-4o Case Study for Critique Data Generation

To illustrate our approach for generating critique data, we present an example case using GPT-4o as the critique generator. The following shows the input prompt, student solution, and the resulting critique generated by GPT-4o.

> **Input Prompt to GPT-4o**
>
> You are a science expert. A student is trying to solve a question, please explain briefly whether his answer is correct or not. Finally, conclude your judgement with 'Conclusion: right/wrong [END]'
> Question:
> How does mitosis contribute to the maintenance and continuity of life in eukaryotic organisms?
> Solution:
> Mitosis plays a crucial role in maintaining and continuing life in eukaryotic organisms by:
> * Growth and Development: Mitosis allows for the production of new cells, enabling organisms to increase in size and develop from a single cell into a multicellular entity.
> * Tissue Repair and Regeneration: When tissues are damaged, mitosis generates new cells to replace the lost or injured ones, facilitating wound healing and tissue regeneration. * Cell Replacement: Mitosis continually replaces old or dying cells with new ones, maintaining the integrity and function of tissues and organs. * Reproduction: In asexual reproduction, mitosis is the primary mechanism by which new individuals are produced. It also plays a role in sexual reproduction by generating gametes (eggs and sperm) that combine to form a new organism.
> Without mitosis, eukaryotic organisms would be unable to grow, repair themselves, or reproduce, ultimately compromising their survival and the continuity of life.

> **GPT-4o Generated Critique**
>
> The student's answer is mostly correct, but there is a significant error in the explanation regarding the role of mitosis in reproduction. While mitosis is indeed essential for growth, tissue repair, and cell replacement, it is not involved in the production of gametes (eggs and sperm) in sexual reproduction. Gametes are produced through a different process called **meiosis**, not mitosis. Mitosis is involved in asexual reproduction for some organisms, but it does not play a direct role in sexual reproduction.
> The rest of the answer correctly highlights the importance of mitosis in growth, tissue repair, and cell replacement, which are vital for the maintenance and continuity of life.
> **Conclusion: wrong [END]**

This example demonstrates how GPT-4o effectively identifies the conceptual error in the student's answer regarding reproduction. The critique correctly points out that meiosis, not mitosis, is responsible for gamete production, while acknowledging the accurate components of the answer. The structured output concludes with a clear judgment, facilitating further computational processing of the critique data. We also used critique data obtained through this instruction to filter and create the WebInstruct-GPT-4o-verified-SFT dataset.

### A.2 GPT-4o Case Study for WebInstruct-GPT-4o-SFT dataset Generation

To illustrate how we generated the WebInstruct-GPT-4o-SFT dataset, we present an example prompt and the corresponding output from GPT-4o.

---

**Input Prompt to GPT-4o**

You are a science expert. You need to answer a given question clearly and conclude your answer with 'Answer: [YOUR ANSWER]
Question: How do you find the derivative of a quadratic function using the Power Rule?

---

**GPT-4o Generated Answer**

To find the derivative of a quadratic function using the Power Rule, follow these steps. A quadratic function is generally in the form:
$f(x) = ax^2 + bx + c$
where $a$, $b$, and $c$ are constants.
The Power Rule for differentiation states that if you have a function $f(x) = x^n$, then its derivative $f'(x)$ is given by:
$f'(x) = nx^{n-1}$
Apply the Power Rule to each term of the quadratic function:
1. **Differentiate $ax^2$:** - Here, $n = 2$, so using the Power Rule: $\frac{d}{dx}(ax^2) = 2ax^{2-1} = 2ax$
2. **Differentiate $bx$:** - Here, $n = 1$, so using the Power Rule: $\frac{d}{dx}(bx) = 1 \cdot bx^{1-1} = b$
3. **Differentiate $c$:** - Since $c$ is a constant, its derivative is 0.
Combine these results to find the derivative of the entire quadratic function:
$f'(x) = 2ax + b$
Thus, the derivative of the quadratic function $f(x) = ax^2 + bx + c$ is:
$f'(x) = 2ax + b$
Answer: The derivative of a quadratic function $f(x) = ax^2 + bx + c$ using the Power Rule is $f'(x) = 2ax + b$.

---

This example demonstrates how GPT-4o provides clear, structured mathematical explanations with step-by-step reasoning. The response includes proper mathematical notation, methodical breakdown of the differentiation process, and a concise final answer. These high-quality responses formed the foundation of our WebInstruct-GPT-4o-SFT dataset.

### A.3 Case Study: Noisy Critique Data

Here we present examples of noisy critique data and analyze their impact on model performance:

### A.3.1 Case 1: Incorrect Critique of Valid Solution

---

**Case 1: Incorrect Critique of Valid Solution**

**Problem**

What is flawed in the counter-proof attempt against the statement that two randomly chosen points on the surface of an N-dimensional unit sphere with a large N are nearly orthogonal? The counter-proof involves selecting a random arc with length in the interval $(0, \pi)$ and placing it on the sphere's surface at a random orientation, claiming that the ends of the arc don't have a high probability of being nearly orthogonal.

**Original Solution**

The flaw in the counter-proof attempt lies in the misunderstanding of the uniform distribution of randomly selected points on the sphere's surface. The original statement assumes that the points are chosen independently and uniformly, whereas the counter-proof proposes a specific distribution by creating a random arc. This distribution does not represent the uniform selection of two points on the sphere.

The Bertrand Paradox, a related concept in probability, demonstrates that different methods of selecting a random chord in a circle can lead to different probabilities for the chord being longer than half the circle's radius. Similarly, in the N-dimensional case, the probability of orthogonality depends on the method of random selection, and the arc method does not refute the original statement about two uniformly chosen points.

To address the issue, one should focus on the probability of two points being nearly orthogonal when they are each chosen independently and uniformly from the sphere's surface, which is the basis of the original statement.

**Incorrect Critique**

Your explanation is on the right track, but it could be clarified further. The key issue with the counter-proof is that it introduces a biased method of selecting points on the sphere. By constructing a random arc and using its endpoints, the counter-proof imposes a specific correlation between the two points. This violates the assumption of independence and uniformity in the original statement.

In high-dimensional spaces (large $N$), the geometry of the sphere leads to the phenomenon where two points chosen independently and uniformly are almost always nearly orthogonal. This is a result of the concentration of measure phenomenon, which is a well-established property in high-dimensional geometry. The counter-proof's method of selecting points via an arc does not reflect this uniform and independent selection process, and thus cannot serve as a valid refutation of the original statement.

**Conclusion: wrong [END]**

**Analysis**

The critique is overly harsh, as the original solution already provides a clear and sufficient answer to the core question.

### A.3.2 Case 2: Missed Errors in Solution

**Case 2: Missed Errors in Solution**

**Problem**

Find the roots of the following polynomial: $\frac{43x^2}{5} + 5x + \frac{28}{5}$.

**Original Solution**

To find the roots of the polynomial, we can use the quadratic formula:
$x = \frac{-b \pm \sqrt{b^2 - 4ac}}{2a}$
where a, b, and c are the coefficients of the polynomial.
In this case, we have:
$a = \frac{43}{5}, b = 5, c = \frac{28}{5}$
Substituting these values into the quadratic formula, we get: $x = \frac{-5 \pm \sqrt{5^2 - 4\left(\frac{43}{5}\right)\left(\frac{28}{5}\right)}}{2\left(\frac{43}{5}\right)}$
Simplifying this expression, we get:
$x = \frac{-5 \pm \sqrt{25 - \frac{43 \cdot 28}{5}}}{2\left(\frac{43}{5}\right)}$ $x = \frac{-5 \pm \sqrt{25 - 238}}{2\left(\frac{43}{5}\right)}$ $x = \frac{-5 \pm \sqrt{-213}}{2\left(\frac{43}{5}\right)}$ $x = \frac{-5 \pm i\sqrt{213}}{2\left(\frac{43}{5}\right)}$ $x = \frac{-5 \pm i\sqrt{4191}}{86}$
Therefore, the roots of the polynomial are $x = \frac{-5 \pm i\sqrt{4191}}{86}$.

**Inadequate Critique**

Let's verify this step by step:
The quadratic formula $\frac{-b \pm \sqrt{b^2 - 4ac}}{2a}$ was correctly applied For the given equation $\frac{43x^2}{5} + 5x + \frac{28}{5}$: $a = \frac{43}{5}$ $b = 5$ $c = \frac{28}{5}$ When we substitute: $b^2 = 25$ $4ac = 4\left(\frac{43}{5}\right)\left(\frac{28}{5}\right) = 238$ $b^2 - 4ac = 25 - 238 = -213$ Therefore: $x = \frac{-5 \pm i\sqrt{213}}{2\left(\frac{43}{5}\right)} = \frac{-5 \pm i\sqrt{4191}}{86}$ The denominator 86 comes from $2 \cdot \frac{43}{5} \cdot 5 = 86$, and $\sqrt{213} \cdot \sqrt{19.67} \approx \sqrt{4191}$
Thus, the answer $x = \frac{-5 \pm i\sqrt{4191}}{86}$ must be correct. Crituque Conclusion: correct
[END]

**Correct Analysis**

The critical calculation error occurs in step 4: The transformation from $\frac{-5 \pm i\sqrt{213}}{2\left(\frac{43}{5}\right)}$ to $\frac{-5 \pm i\sqrt{4191}}{86}$ is incorrect. 1) While the denominator's 5 was properly handled ($2 \cdot \frac{43}{5} = \frac{86}{5}$) 2) It ignored that this coefficient 5 must affect the numerator as well The correct simplification should be: $\frac{-5 \pm i\sqrt{213}}{2\left(\frac{43}{5}\right)} = \frac{-25 \pm 5i\sqrt{213}}{86}$.

## A.4 Detailed Analysis of Self-Critique Methods

We experimented with two self-critique approaches during inference:

1. **Single-pass self-critique**: The model generates and critiques its solution within one inference pass, and regenerates if errors are detected.

2. **Two-stage self-critique**: The model separately evaluates its solutions in a dedicated critique step and iteratively regenerates (up to 8 attempts) until a satisfactory solution is found.

Table 11 shows the comparative performance of these methods across various temperature settings.

| Method | Temperature | MATH | Minerva-Math |
|---|---|---|---|
| Direct inference | 0.0 | 80.2 | 42.3 |
|  | 0.1 | 78.8 | 38.9 |
|  | 0.3 | 77.5 | 37.7 |
|  | 0.6 | 75.2 | 34.1 |
| Single-pass self-critique | 0.1 | 77.2 | 36.7 |
|  | 0.3 | 76.1 | 35.2 |
|  | 0.6 | 73.5 | 34.3 |
| Two-stage self-critique | 0.1 | 77.9 | 38.2 |
|  | 0.3 | 75.8 | 35.4 |
|  | 0.6 | 74.6 | 34.6 |

Table 11: Comparison of inference methods across various temperature settings.

Our analysis revealed several issues that limit the effectiveness of self-critique:

1. **Inconsistent critique standards**: The model often applies inconsistent criteria when evaluating its own work, leading to either missed errors (false negatives) or incorrectly flagging valid solutions as problematic (false positives).

2. **Temperature sensitivity**: Higher temperatures introduce variability that compounds across iterations, making the overall process less stable. The single-pass method drops from 77.2% to 73.5% on MATH as temperature increases from 0.1 to 0.6, with similar trends on Minerva-Math.

3. **Regeneration inefficiency**: Even when errors are correctly identified, the model often struggles to effectively address the specific issues in subsequent regeneration attempts, sometimes introducing new errors while fixing others.

4. **Computational overhead**: The iterative nature of self-critique significantly increases inference time and computational cost, with diminishing returns on performance.

The two-stage method performs slightly better than the single-pass approach but still underperforms relative to direct inference. This suggests that while our model benefits from critique-based training, applying self-critique during inference creates additional complexity that the model struggles to navigate effectively.

### A.4.1 Inference Method Prompts

We present the prompt templates used in our different inference approaches:

**Direct Inference Template**

> **Direct Inference Prompt Template**
>
> Please reason step by step, and put your final answer within
> boxed.
> Question: [Problem text here]
> Answer: Let's solve this step by step: [Solution steps] Therefore, the final answer is $\boxed{ANSWER}$.

**Single-pass Self-critique Template**

---

**Single-pass Self-critique Prompt Template**

Please reason step by step to solve this problem and then critique your solution. If any errors are found, provide a corrected solution. Please put your final answer within $\square$.
Question: [Problem text here]
Answer: Let's solve this first: [Initial solution steps] Therefore, my initial answer is $\boxed{ANSWER}$.
Critique: [Critique points]
[If errors found: Based on my critique, let me provide a corrected solution: Corrected solution: ...]

---

**Two-stage Self-critique Template**

---

**Two-stage Self-critique Process**

**Stage 1 (Solution Generation):**

> Please reason step by step, and put your final answer within boxed.
> Question: [Problem text here]
> Answer: Let's solve this step by step: [Solution steps] Therefore, the final answer is $\boxed{ANSWER}$.

**Stage 2 (Critique):**

> Please critique whether the following solution to the question is correct.
> Question: [Problem text here] Solution: [Previous solution]
> Critique: 1. [Critique point 1] 2. [Critique point 2] ...
> Critique Conclusion: Correct/Incorrect

If the conclusion is "Incorrect", the process returns to Stage 1 for a new solution attempt. This iterative process continues until either:

- The critique conclusion becomes "Correct", indicating a satisfactory solution has been found, or

- The maximum number of iterations (8) is reached, in which case the last generated solution is used as the final answer.

The complete process can be represented as:

> For i in range(1, 9): 1. Generate solution (Stage 1) 2. Critique solution (Stage 2) 3. If critique conclusion is "Correct": Return current solution and exit 4. If i == 8: Return current solution 5. Continue to next iteration

---

### A.4.2 Observed Issues in Self-Critique Methods

---

**Common Issues with Self-Critique Methods**

Our analysis revealed several systematic issues with self-critique approaches:

*Inconsistent Self-Evaluation*
We observed the following patterns of inconsistency in self-evaluation:

- **Over-lenient Evaluation**: Cases where the model failed to identify clear logical errors in its own reasoning
- **False Negatives**: Instances where correct solutions were incorrectly flagged as erroneous

*Specific Failure Cases*
Here we document specific examples where self-critique methods failed:

---

**Example 1: Over-lenient Evaluation**

**Problem**

Chandra has four bowls. Each one is a different color (red, blue, yellow, green). She also has exactly one glass the same color as each bowl. If she chooses a bowl and a glass from the cupboard, how many pairings are possible? One such pairing is a blue bowl and a yellow glass.

**Original Solution**

To solve this, we need to determine the total number of possible pairings of one bowl and one glass when there are four bowls and four glasses, each uniquely paired by color.
Reasoning: 1. Chandra has **4 bowls** (red, blue, yellow, green) and **4 glasses** (red, blue, yellow, green). 2. Each bowl matches exactly one glass by color. For example: - The red bowl matches with the red glass - The blue bowl matches with the blue glass - The yellow bowl matches with the yellow glass - The green bowl matches with the green glass 3. Since each bowl must pair with its corresponding glass, there is only **1 valid pairing** (a red bowl with a red glass, etc.).

**Self-Critique**

The question asks for the number of possible pairings, but based on the problem setup, only one valid pairing exists. The reasoning above is correct.
**Conclusion: Correct [END]**

## Example 2: False Error Detection

### Problem

Chandra has four bowls. Each one is a different color (red, blue, yellow, green). She also has exactly one glass the same color as each bowl. If she chooses a bowl and a glass from the cupboard, how many pairings are possible? One such pairing is a blue bowl and a yellow glass.

### Original Solution

To solve this problem, we need to calculate the total number of possible pairings of bowls and glasses. Each bowl can be paired with any of the 4 glasses.
Here are the steps:
1. Chandra has 4 bowls and 4 glasses. 2. Each bowl can be paired with any one of the 4 glasses. 3. Therefore, the total number of pairings is calculated as the product of the number of bowls and the number of glasses:

$$4 \times 4 = 16$$

### Self-Critique

The reasoning provided is incorrect. While the calculation of the total number of pairings as $4 \times 4 = 16$ is mathematically correct, the explanation is flawed. The problem does not state that each bowl must be paired with a glass of a different color. Therefore, the assumption that each bowl can only be paired with a glass of the same color is invalid.
Conclusion: **Critique Conclusion: Incorrect**

