# OpenReview forum: "Critique Fine-Tuning: Learning to Critique is More Effective than Learning to Imitate"
_colmweb.org/COLM/2025/Conference — COLM 2025_

### Official Review · Reviewer_G4No · 2025-05-11

**Rating:** 7
**Confidence:** 4
**Ethics Flag:** 1

**Summary:**

This paper introduces Critique Fine-Tuning (CFT), a novel supervised fine-tuning strategy for large language models (LLMs) aimed at enhancing their reasoning and analytical capabilities. CFT trains models to generate critiques given a query and a noisy response, using GPT-4o-generated critique data of the form ([query, noisy response], critique). Through extensive experiments, the authors demonstrate that CFT significantly improves performance across multiple reasoning benchmarks. The method shows robustness to imperfect data, different solution sources, and teacher models. The paper is clearly written and provides ablation studies to examine the contributions of various components.

**Reasons To Accept:**

1. The paper presents a simple but effective fine-tuning strategy that can be readily applied to existing LLMs.

2. Empirical results show consistent improvements over baselines across diverse settings.

3. The proposed CFT approach is robust to noisy data and model variations, making it broadly applicable.

4. The authors provide meaningful ablations and experimental validations to support their claims.

**Reasons To Reject:**

1. The core idea—training with critique data—may be perceived as intuitive or incremental, potentially lacking strong novelty.

2. The connection between the CFT objective and actual model behavior post-fine-tuning is underexplored; more discussion or analysis is needed to justify why this approach leads to better reasoning.

---

> ### Author Response · Authors · 2025-06-01
>
> We thank the reviewer for the positive evaluation and constructive suggestions.
>
> ## On Novelty and Conceptual Contribution
> Our approach is a paradigm shift rather than a simple change of training data. By utilizing critique-based data, CFT unlocks the model’s reasoning potential and effectively mitigates the overfitting risk associated with SFT. Instead of biasing the model toward a narrow set of reference answers, CFT exposes it to teacher critiques across diverse candidate responses, allowing the model to learn from various perspectives and error patterns. This fosters deeper understanding and more robust analytical capabilities.
>
> ## Connection Between CFT Objective and Improved Reasoning
> Regarding the connection between the CFT objective and improved reasoning performance, we offer a theoretical framework to explain this relationship:
>
> The fundamental difference between SFT and CFT lies in their information structure during learning. Traditional SFT requires models to learn a direct mapping from questions (x) to answers (y), without intermediate reasoning steps. This resembles asking students to produce final answers without showing their work.
>
> CFT, however, operates on richer information: it trains on triples consisting of a question (x), an initial answer attempt (y'), and a detailed critique (c) of that attempt. This approach parallels educational practices where students learn by analyzing worked examples with instructor feedback.
>
> From an information-theoretic standpoint, CFT provides more structured guidance for the learning process. The model must process not only the question and correct answer, but also understand common error patterns, solution strategies, and verification techniques through the critique process. This conditioning on additional context (the initial answer) provides a more constrained learning objective that focuses specifically on reasoning failures and their corrections. This view is also supported by findings in "Critique-Guided Distillation: Improving Supervised Fine-tuning via Better Distillation," which demonstrates that critique-driven supervision provides richer learning signals and leads to better generalization.
>
> Our experimental results validate this theoretical advantage - models trained with CFT consistently outperform their SFT counterparts across diverse benchmarks, demonstrating more robust reasoning capabilities with significantly less training data. This efficiency suggests that learning from critiques enables more effective knowledge transfer than simply imitating correct answers.
>
> We will expand this analysis in our revised manuscript to provide a clearer theoretical justification for CFT's effectiveness.

---

> > ### Comment · Reviewer_G4No · 2025-06-07
> >
> > Thank you for the authors’ response. I remain supportive of the paper and will retain my positive evaluation.”

---

### Official Review · Reviewer_tGLF · 2025-05-11

**Rating:** 6
**Confidence:** 4
**Ethics Flag:** 1

**Summary:**

This paper introduces Critique Fine-Tuning (CFT), a novel approach to training language models that departs from traditional Supervised Fine-Tuning (SFT). Instead of directly imitating responses, CFT trains models to critique and analyze generated responses, focusing on identifying and correcting errors in reasoning. The authors demonstrate that this approach consistently outperforms SFT by 4-10% on mathematical reasoning benchmarks, achieves comparable performance to resource-intensive reinforcement learning (RL) methods using significantly fewer training examples and compute resources, and generalizes effectively to broader STEM domains. Notably, CFT-trained models exhibit strong instruction-following capabilities without explicit instruction tuning, challenging conventional assumptions about the necessity of imitation-based training. The core idea is that by explicitly teaching models to identify and critique incorrect reasoning, their reasoning and generalization capabilities can be significantly enhanced. The authors achieve this by fine-tuning models on a dataset of question-response pairs, where the response is a critique of the model's initial answer. This critique is generated by a powerful language model, GPT-4o, and includes an assessment of the answer's correctness, identification of errors, and a corrected solution. The experiments are thorough, comparing CFT with various SFT methods and RL-based approaches, and the results provide strong empirical support for the effectiveness of CFT. The paper also explores the limitations of CFT, including the impact of noisy critique data and the challenges of self-critique mechanisms.

**Questions To Authors:**

Please check the "reasons to reject".

**Reasons To Accept:**

This paper presents a compelling case for Critique Fine-Tuning (CFT) as an alternative to traditional Supervised Fine-Tuning (SFT), backed by strong empirical evidence and thoughtful analysis. The experimental evaluation is comprehensive, covering a wide range of open- and closed-source LLMs across multiple mathematical reasoning benchmarks. Notably, the authors demonstrate that models trained with CFT on relatively small datasets can match or surpass the performance of models trained with significantly larger datasets or more compute-intensive methods such as reinforcement learning. This indicates the method's high data efficiency and practical relevance. Additionally, the work challenges conventional views on instruction tuning by showing that CFT-trained models achieve strong instruction-following capabilities without explicit imitation-based training, highlighting the broader applicability and impact of critique-based learning.

**Reasons To Reject:**

1. **Unclear Inference Procedure**: While the training setup is reasonably well described, using question and noisy response pairs as inputs and critiques as outputs, the testing procedure remains ambiguous. Specifically, it is unclear how the model is expected to generate final answers at inference time. If the critique merely assesses an initial (possibly incorrect) response without producing a corrected answer, it raises the question of how the model ultimately arrives at a valid final output. The paper does not explain whether critique outputs are further utilized to refine or replace the initial response, leaving a critical gap in understanding the method’s practical utility.

2. **Insufficient Positioning Against Prior Work**: The idea of self-critique or language-model-generated feedback has been explored in several recent works [1,2,3], including multi-agent critique pipelines and feedback-based self-improvement. However, the paper lacks an adequate comparison to these methods, either theoretically or empirically. Without situating CFT more clearly in the context of this growing literature, it is difficult to assess its novelty or advantages over existing critique-based training frameworks.

3. **Lack of Analysis in High-Quality SFT Regimes**: One of the motivating claims is that CFT can be more efficient or effective than conventional SFT, particularly when training data is limited or noisy. However, the paper does not sufficiently explore how CFT compares to high-quality, clean SFT scenarios. If critiques are generated from potentially flawed or noisy model outputs, it is unclear whether GPT-4o feedback is beneficial or harmful when the baseline data already contains correct answers. A deeper analysis or controlled experiments in such settings would help clarify the robustness and boundaries of CFT’s applicability.



[1] Lan, T., Zhang, W., Lyu, C., Li, S., Xu, C., Huang, H., ... & Chen, K. (2024). Training language models to critique with multi-agent feedback. arXiv preprint arXiv:2410.15287.
[2] Hu, C., Hu, Y., Cao, H., Xiao, T., & Zhu, J. (2024). Teaching Language Models to Self-Improve by Learning from Language Feedback. arXiv preprint arXiv:2406.07168.
[3] Feng, T., Wang, Z., & Sun, J. (2023). Citing: Large language models create curriculum for instruction tuning. arXiv preprint arXiv:2310.02527.

---

> ### Author Response · Authors · 2025-06-01
>
> Thank you for your careful reading and thoughtful comments. We address your main concerns as follows:
>
> ## Inference Procedure Clarity
>
> At inference time, as described in Appendix A.4.1, we use the same prompt and procedure as standard SFT baselines: the model is directly prompted to generate a final answer, not a critique. Interestingly, even though the CFT training objective differs from the evaluation task, models fine-tuned with CFT demonstrate superior performance across benchmarks. This suggests that learning to critique diverse solutions helps the model internalize better reasoning patterns, which transfer to direct answer generation. This phenomenon is similar to how students deepen their understanding by grading others' work and subsequently perform better in exams.
>
> To clarify this important point, we will add more details about the inference procedure in the main text of the revised manuscript.
>
> ## Positioning Against Prior Work
>
> Lan et al. (2024) propose MultiCritique, a pipeline that aggregates critiques from multiple agents to enhance the critique ability of language models. Their approach involves both supervised fine-tuning and reinforcement learning stages, utilizing multi-agent feedback to improve critique quality. In contrast, our Critique Fine-Tuning (CFT) method trains a single model to critique noisy responses using GPT-4o-generated critiques, eliminating the need for multi-agent aggregation and reinforcement learning, thereby simplifying the training process.
>
> Hu et al. (2024) introduce Self-Refinement Tuning (SRT), where a base model generates initial responses that are critiqued and refined by a more advanced model, enabling the base model to learn from this feedback. This process creates a feedback loop that promotes model improvement. While SRT relies on a teacher-student framework and iterative self-critique, our CFT approach trains a single model end-to-end to critique and improve upon noisy responses, reducing dependency on more advanced models for feedback and enabling the model to directly generate improved answers without iterative refinement.
>
> Feng et al. (2023) present CITING, where a teacher model creates rubrics and revisions to guide the instruction tuning of a student model. This method emphasizes curriculum learning through rubric-based feedback. In contrast, our CFT approach focuses on training models to critique noisy responses directly, without the need for rubric generation or a separate teacher model, streamlining the training pipeline.
>
> We will clarify how CFT is distinct in its paradigm and practical implementation, and discuss its relative strengths and novelty compared to these frameworks.
>
> ## Comparison With High-Quality SFT
>
> We have included experiments comparing CFT with high-quality SFT data, as shown in Table 5:
>
> | Task | MetaMathQA |  | NuminaMath |  | WebInstruct |  |
> |------|------------|-------------|-------------|-------------|-------------|-------------|
> |      | SFT        | CFT         | SFT         | CFT         | SFT         | CFT         |
> | MATH | 57.5       | 74.4        | 70.8        | 74.2        | 59.0        | 80.2        |
> | Minerva-Math | 23.9 | 42.3      | 28.3        | 32.8        | 13.2        | 42.3        |
> | GSM8K | 79.5      | 85.7        | 88.3        | 89.1        | 77.4        | 90.9        |
> | OlympiadBench | 20.0 | 36.4     | 36.3        | 37.2        | 19.9        | 41.6        |
> | AIME24 | 6.7      | 23.3        | 10.0        | 23.3        | 3.3         | 20.0        |
> | AMC23 | 37.5      | 57.5        | 50.0        | 62.5        | 37.5        | 67.5        |
> | AVG | 37.5       | 53.3        | 47.3        | 53.2        | 35.1        | 57.1        |
>
> Results show that CFT remains more effective, even when compared to these strong SFT baselines. Additionally, Table 3 has shown comparisons with numerous high-quality SFT baselines, including NuminaMath-7B-CoT, Eurus-2-SFT, rStar-Math@Greedy, and AceMath-Qwen2.5-Math:
>
> | Model | #Data | MATH | GPQA | TheoremQA | MMLU-Pro | OlympiadBench | AIME24 | AMC23 | AVG |
> |-------|-------|------|------|-----------|----------|---------------|--------|-------|-----|
> | Qwen2.5-Math-Base | 0 | 55.4 | 31.0 | 37.4 | 39.3 | 16.1 | 10.0 | 40.0 | 32.7 |
> | Eurus-2-SFT | 230K | 62.4 | 32.1 | 38.0 | 44.2 | 29.8 | 3.3 | 30.1 | 34.3 |
> | rStar-Math@Greedy | 747K | 78.4 | - | - | - | 47.1 | 26.7 | 47.5 | - |
> | AceMath-Qwen2.5-Math | 2.3M | 83.1 | 26.1 | 24.6 | 48.1 | 42.2 | 16.7 | 60.0 | 43.0 |
> | Qwen2.5-Math-7B-Instruct | 2.5M | 83.6 | 31.1 | 37.0 | 39.5 | 41.6 | 16.7 | 62.5 | 44.6 |
> | Qwen2.5-Math-7B-CFT | 50K | 80.2 | 39.4 | 40.4 | 47.5 | 41.6 | 20.0 | 67.5 | 48.1 |
>
> These results demonstrate CFT's advantages over high-quality SFT approaches, while using significantly fewer training examples (50K vs. millions).
>
> We will incorporate these improvements and clarifications into our revised submission.

---

> > ### Comment · Reviewer_tGLF · 2025-06-04
> >
> > I appreciate the authors' response to my concern. For the newly added comparison results between SFT and CFT, is there a reason why the high-quality SFT data yields much lower performance than CFT?

---

> > > ### Author Response · Authors · 2025-06-05
> > >
> > > Thank you for your question. Just to clarify, the two tables you mentioned are not newly added in the revised version; they were already included in the original submission.
> > >
> > > To ensure a fair comparison in Table 5, we sampled the same number of examples (50K) for both SFT and CFT. Despite using high-quality data, SFT still underperforms compared to CFT. We believe this is because SFT tends to overfit when trained on small-scale data. In contrast, CFT encourages the model to analyze errors and reason critically by exposing it to diverse perspectives and error types, leading to better generalization and more robust reasoning abilities.

---

### Official Review · Reviewer_Yors · 2025-05-14

**Rating:** 8
**Confidence:** 3
**Ethics Flag:** 1

**Summary:**

This paper proposes a novel method for training large language models (LLMs) using synthetic data derived from critiques of instruction-tuning datasets, rather than imitating high-quality responses. The proposed Critique Fine-Tuning (CFT) framework is shown to be both data- and compute-efficient, requiring significantly fewer resources than traditional supervised fine-tuning (SFT) while consistently outperforming SFT-based models across a wide range of benchmarks.

**Reasons To Accept:**

1. The paper is clearly written and well-organized. The appendix provides detailed insights into experimental settings and analyses.
2. The method is straightforward and easy to implement, yet novel and conceptually elegant.
3. The experiments are thorough and diverse. Ablation studies further support the robustness and generality of the approach.
4. The proposed method is highly efficient: it achieves strong performance using only 50K training examples and 1 hour of training on 8×H100 GPUs, far less than competing methods.

**Reasons To Reject:**

Most experiments are conducted on English datasets. Apart from MMLU-Pro, the method’s effectiveness in multilingual or non-English settings remains unclear.

---

> ### Author Response · Authors · 2025-06-01
>
> We thank the reviewer for the positive and encouraging feedback. We acknowledge the limitations regarding multilingual evaluation. In future work, we will conduct CFT experiments on other language versions of WebInstruct and then evaluate the performance on other language tasks in benchmarks like MMLU-Pro, Math, to explore multilingual capabilities.

---

> > ### Comment · Reviewer_Yors · 2025-06-07
> >
> > Thank you for your clarification. I look forward to seeing your future work!

---

### Official Review · Reviewer_XEXb · 2025-05-23

**Rating:** 5
**Confidence:** 3
**Ethics Flag:** 1

**Summary:**

This paper proposes Critique Fine-Tuning (CFT) as a compelling alternative to standard Supervised Fine-Tuning (SFT) for reasoning tasks in language models. Instead of training models to imitate correct answers, CFT trains them to critique noisy responses, thereby fostering deeper reasoning and analytical capabilities. The authors construct multiple critique datasets (e.g., WebInstruct, MetaMath, NuminaMath) where critiques are generated by GPT-4o. Empirical results show CFT outperforms strong SFT baselines by 4–10% on various math reasoning benchmarks and even competes with compute-intensive RL-based approaches like SimpleRL. Ablations demonstrate robustness across data sources and critique models. The work is well-motivated, clearly presented, and makes a strong case for critique-based learning as a scalable, data-efficient training paradigm.

**Reasons To Accept:**

- The paper introduces a learning paradigm in which language models are trained to critique rather than imitate, drawing a clear and compelling motivation from human learning processes.
- The proposed Critique-Fine-Tuning (CFT) method achieves strong empirical performance, significantly outperforming supervised fine-tuning (SFT) on multiple benchmarks—for instance, achieving a +6.7% gain on Qwen2.5-Math-7B—and performing comparably to reinforcement learning approaches like SimpleRL while being 140 times more compute-efficient.
- The method demonstrates remarkable data efficiency, with models trained on only 50K critique examples matching or even surpassing the performance of models trained with millions of supervised examples.
- The approach shows robust generalization capabilities across different datasets, teacher models, and reasoning domains. It improves both reasoning accuracy and general instruction-following ability, achieving, for example, a +1.0 improvement on MT-Bench over the best SFT baseline.
- The paper is clearly written and well-organized, featuring thorough ablation studies, strong conceptual motivation, and effective visualizations that together enhance the clarity and persuasiveness of its core contributions.

**Reasons To Reject:**

- The paper demonstrates strong empirical performance, but it does not provide a rigorous theoretical explanation for why critique-based learning enhances reasoning capabilities. This absence of formal justification weakens the conceptual grounding of the work.
- The model relies on critiques generated by GPT-4o, which introduces a dependency on potentially biased or inaccurate supervision. Manual inspection reveals that around 20% of the critiques are flawed, raising concerns about the reliability of the training signal.
- The evaluation focuses predominantly on math-related benchmarks, making it unclear whether the proposed method generalizes well to non-STEM or open-ended domains. Additional experiments are needed to establish the broader applicability of the approach.
- Although the paper briefly mentions self-critique mechanisms, it does not explore in depth why these methods fail or how they might be improved. As a result, the potential for self-improving critique-based training remains insufficiently addressed.
- More related work and baselines should be discussed, e.g., [1][2].

[1] Enhancing LLM Reasoning via Critique Models with Test-Time and Training-Time Supervision
[2] Critic-CoT: Boosting the reasoning abilities of large language model via Chain-of-thoughts Critic

---

> ### Author Response · Authors · 2025-06-01
> **Response to Reviewer XEXb**
>
> We sincerely thank Reviewer XEXb for the thoughtful and constructive feedback. Below, we address each concern point-by-point.
>
> ## Rigorous Theoretical Explanation for CFT's Effectiveness
>
> We can explain why CFT outperforms SFT from a Bayesian perspective that SFT and CFT differ fundamentally in the amount of information available for learning.
>
> * **SFT** learns the mapping P(y|x): given only the input x, the model is expected to produce the correct answer y. This setting is analogous to inferring an output solely from a prior, without any additional evidence.
> * **CFT**, in contrast, learns P(c|x, y'): given the input x and the model's initial answer y', the model is trained to generate a critique c. Here, y' provides an explicit instantiation of the model's prior reasoning, and the critique c serves as feedback or "evidence" that highlights errors or gaps in reasoning.
>
> In Bayesian terms, conditioning on more evidence (here, y') reduces uncertainty in the model's output distribution. The critique c can be seen as an update based on the "observation" of y', sharpening the model's internal representations and focusing learning on specific reasoning failures.
>
> Formally, the conditional entropy H(C|X, Y') is always less than or equal to H(C|X), meaning that the output distribution in CFT is more constrained and easier to learn than in SFT. This additional conditioning enables CFT to develop more precise and diagnostic internal representations, which empirically leads to improved reasoning performance.
>
> This view is also supported by findings in "Critique-Guided Distillation: Improving Supervised Fine-tuning via Better Distillation," which demonstrates that critique-driven supervision provides richer learning signals and leads to better generalization.
>
> ## Reliance on GPT-4o Critiques and Data Quality Concerns
>
> We explore the impact of teacher model quality on CFT effectiveness in Table 7, where we compare different critique models:
>
> | Task | SFT | GPT-4o-mini-CFT | GPT-4o-1120-CFT |
> |------|-----|-----------------|-----------------|
> | MATH | 62.0 | 73.9 | 80.2 |
> | Minerva-Math | 12.5 | 36.4 | 42.3 |
> | GSM8K | 78.8 | 84.5 | 90.9 |
> | OlympiadBench | 22.1 | 35.1 | 41.6 |
> | AIME24 | 16.7 | 20.0 | 20.0 |
> | AMC23 | 50.0 | 62.5 | 67.5 |
> | AVG | 40.4 | 52.0 | 57.1 |
>
> While stronger teachers (GPT-4o-1120) yield better results, even weaker teachers (GPT-4o-mini) significantly outperform SFT baselines. Additionally, we conducted experiments using WebInstruct-verified and DeepScaler datasets, which contain verifiable data to validate critique accuracy. After filtering out noisy data, performance remained largely unchanged, suggesting that the model learns the pattern of critical evaluation rather than depending on the precise accuracy of individual critiques. We will expand this discussion in our revised manuscript.
>
> ## Evaluation Beyond Math/Reasoning Benchmarks
>
> We evaluate performance on general instruction-following tasks in Table 8:
>
> | Model | IF_Eval |  | MT-Bench |
> |-------|--------|--------|----------|
> |  | strict | loose |  |
> | Qwen2.5-Math-7B | 0.266 | 0.291 | 4.79 |
> | Qwen2.5-Math-7B-Instruct | 0.333 | 0.345 | 5.49 |
> | Qwen2.5-Math-7B-SFT | 0.315 | 0.330 | 5.23 |
> | Qwen2.5-Math-7B-verified-SFT | 0.328 | 0.341 | 5.41 |
> | Qwen2.5-Math-7B-GPT4o-SFT | 0.325 | 0.343 | 5.38 |
> | Qwen2.5-Math-7B-CFT | 0.335 | 0.362 | 6.49 |
>
> Results show that CFT, despite being trained on WebInstruct, effectively improves general instruction-following capabilities, significantly outperforming SFT baselines on both IF_Eval and MT-Bench metrics.
>
> ## Analysis of Self-Critique Mechanisms
>
> Regarding self-critique mechanisms, our Table 11 experiments demonstrate that models trained with CFT do not further benefit from self-critique methods during inference:
>
> | Method | Temperature | MATH | Minerva-Math |
> |--------|-------------|------|--------------|
> | Direct inference | 0.0 | 80.2 | 42.3 |
> |  | 0.1 | 78.8 | 38.9 |
> |  | 0.3 | 77.5 | 37.7 |
> |  | 0.6 | 75.2 | 34.1 |
> | Single-pass self-critique | 0.1 | 77.2 | 36.7 |
> |  | 0.3 | 76.1 | 35.2 |
> |  | 0.6 | 73.5 | 34.3 |
> | Two-stage self-critique | 0.1 | 77.9 | 38.2 |
> |  | 0.3 | 75.8 | 35.4 |
> |  | 0.6 | 74.6 | 34.6 |
>
> We hypothesize that during CFT training, models already internalize critical reasoning abilities. Therefore, additional self-critique at inference time does not yield improvements beyond the model's existing capabilities. Direct inference with lower temperature settings consistently yields the best performance.
>
> We appreciate the additional paper references and will incorporate them into our related work discussion in the revised manuscript.

---

> > ### Author Response · Authors · 2025-06-01
> >
> > ## Related work to be discussed
> >
> > Comparison with "Enhancing LLM Reasoning via Critique Models with Test-Time and Training-Time Supervision (Xi et al. 2024)": Xi et al. propose a two-model framework where a separate critique model provides step-level feedback to supervise a reasoning model during both training and inference. Their approach relies on the AutoMathCritique dataset, comprising 76,321 samples with step-level critiques. In contrast, our Critique Fine-Tuning (CFT) method trains a single model to critique noisy responses, eliminating the need for a separate critique model and demonstrating superior data efficiency with only 50K examples.
> >
> > Comparison with "Critic-CoT: Boosting the reasoning abilities of large language model via Chain-of-thoughts Critic (Zheng et al. 2024)": Zheng et al. introduce Critic-CoT, which enhances reasoning by integrating a chain-of-thought (CoT) critic that provides step-wise feedback. Their method emphasizes System-2-like analytic reasoning and utilizes automatically constructed distant-supervision data. Our CFT approach differs by focusing on training models to critique noisy responses without the need for step-wise CoT supervision, offering a more streamlined training process.
> >
> > We will incorporate discussions of these works into our revised manuscript to provide a more comprehensive overview of related research.

---

> > > ### Author Response · Authors · 2025-06-05
> > > **Looking Forward to Your Feedback**
> > >
> > > Hi Reviewer XEXb,
> > >
> > > We’d love to hear your thoughts or feedback on our responses. Please feel free to share any additional comments or questions you might have!

---

> > > > ### Author Response · Authors · 2025-06-07
> > > > **Follow-up on Our Response**
> > > >
> > > > Does our response address your concerns? We’d love for you to join the discussion.

---

> > > > > ### Author Response · Authors · 2025-06-10
> > > > >
> > > > > We have provided detailed responses to your comments and would greatly appreciate your engagement in the discussion. Given the significance of the concerns raised and the impact of your review, we believe a follow-up would be important to ensure a constructive and complete review process.

---

### Decision · Program_Chairs · 2025-07-08

**Decision:**

Accept

**Comment:**

This paper proposed a novel fine-tuning method called Critique Fine-Tuning (CFT), which is more effective than the popular SFT for reasoning tasks as demonstrated in the experiments. Reviewers show unanimous support to the acceptance expect for XEXb. The authors have prepared careful responses to the questions raised by XEXb. I believe it is ready for publication after the authors incorporate the rebuttal responses into the final version. Therefore, I recommend Accept for this work.